# Regulating the Electron Depletion Layer of Au/V_2_O_5_/Ag Thin Film Sensor for Breath Acetone as Potential Volatile Biomarker

**DOI:** 10.3390/nano13081372

**Published:** 2023-04-14

**Authors:** Bader Mohammed Alghamdi, Nawaf Mutab Alharbi, Ibrahim Olanrewaju Alade, Badriah Sultan, Mohammed Mansour Aburuzaizah, Turki N. Baroud, Qasem A. Drmosh

**Affiliations:** 1Materials Science and Engineering Department, King Fahd University of Petroleum & Minerals, Dhahran 31261, Saudi Arabia; s201935230@kfupm.edu.sa (B.M.A.); s201923750@kfupm.edu.sa (N.M.A.); s201932270@kfupm.edu.sa (M.M.A.); turkibaroud@kfupm.edu.sa (T.N.B.); 2Physics Department, King Fahd University of Petroleum & Minerals, Dhahran 31261, Saudi Arabia; ialade@kfupm.edu.sa; 3Department of Physics, King Abdulaziz University, Jeddah 21589, Saudi Arabia; bsultan0007@stu.kau.edu.sa; 4Interdisciplinary Research Centre for Hydrogen and Energy Storage (HES), King Fahd University of Petroleum and Minerals (KFUPM), Dhahran 31261, Saudi Arabia

**Keywords:** acetone, gas sensors, thin films, exhaled breath, biomarker, depletion layer

## Abstract

Human exhaled breath has been utilized to identify biomarkers for diseases such as diabetes and cancer. The existence of these illnesses is indicated by a rise in the level of acetone in the breath. The development of sensing devices capable of identifying the onset of lung cancer or diabetes is critical for the successful monitoring and treatment of these diseases. The goal of this research is to prepare a novel breath acetone sensor made of Ag NPs/V_2_O_5_ thin film/Au NPs by combining DC/RF sputtering and post-annealing as synthesis methods. The produced material was characterized using X-ray diffraction (XRD), UV-Vis, Raman, and atomic force microscopy (AFM). The results revealed that the sensitivity to 50 ppm acetone of the Ag NPs/V_2_O_5_ thin film/Au NPs sensor was 96%, which is nearly twice and four times greater than the sensitivity of Ag NPs/V_2_O_5_ and pristine V_2_O_5_, respectively. This increase in sensitivity can be attributed to the engineering of the depletion layer of V_2_O_5_ through the double activation of the V_2_O_5_ thin films with uniform distribution of Au and Ag NPs that have different work function values.

## 1. Introduction

Human exhalation contains crucial health information such as dietary patterns, disease symptoms, and cell metabolism. A few gaseous components, particularly volatile organic compounds (VOCs), which make up part of our exhaled breath, are known to reveal human health status and can be used to diagnose several diseases [1,2,3,4,5]. Exhaled breath contains numerous gases—for example, 78% nitrogen, 16% oxygen, and 4% carbon dioxide—and hundreds of trace gaseous components including VOCs. In 1971, Linus Pauling utilized gas chromatography for the first time to detect 250 components in human breath [6]. Scientists can currently identify over 300 VOCs and other particles in human breath [7]. Furthermore, many exhaled volatile chemicals can be employed as biological indicators for early illness diagnostics [8]. For example, the biomarker isoprene present in exhaled breath can be used to detect liver illness [9]. Cancer is difficult to identify at the initial stage due to many cancers having no evident symptoms other than elevated VOC levels [10]. As a result, for many cancers, an early diagnosis may have a significant impact on the treatment process. When colon cancer is detected early, more than 90% of patients live for five years or more [11]. Therefore, there is a never-ending search for low-cost, simple-to-use, rapid, and sensitive diagnostic procedures that may be employed on a large section of the population regularly [12]. Exhaled breath is a highly effective tool for detecting various diseases early [13]. The primary reasons for this are the significant advantages it provides, such as the quick, non-invasive, and real-time diagnosis of complicated disorders like cancer and acute infections [14]. This approach has several advantages over traditional diagnostic approaches, including painless operation and sample collection without the assistance of qualified medical personnel [15].

One of the volatile chemicals found in exhaled breath is acetone [16]. The acetone concentration in healthy persons’ exhaled breath is approximately 11.59 ppb, whereas it is approximately 44.68 ppb in diabetes patients [16]. Furthermore, the highest acetone concentration in lung cancer patients’ exhaled breath is about 1 ppm, with an average concentration of 400 ppb, which is more than double the value found in healthy people [17,18,19,20]. As a result, this phenomenon provides an opportunity to use the association between acetone and diabetes as an early detection approach for diabetes. Among other exhaled breath markers (EBM) methodologies such as chromatography, proto-transfer reaction-mass spectrometry, laser spectroscopy, and selected ion flow tube-mass spectrometry, chemo-resistive metal oxide-based gas sensors (MGSs) are receiving considerable attention in the pursuit of reliable acetone sensing technology [21].

MGSs are receiving increasing attention as a result of their favorable characteristics like portability, ease of integration in a chip, simple fabrication method, low fabrication cost, high stability, and reusability [22]. In recent years, many MGSs for acetone detection have been developed. Wang et al. [23], for example, used a one-pot hydrothermal technique to create porous TiO_2_ nanoparticles (NPs) for acetone sensing. For optimal sensing performance, 2 mol% Ag was shown to be optimal. The sensor has a quick response, short recovery time (11 s/14 s), and long-term stability (30 days). Yoo et al. evaluated the performance of ZnO NPs doped with different dopants, including Al, Co, and Cu [24]. Their research showed that Al-doped Zn NPs function better, with a maximum response of 11.8 at 500 °C and sensitivity to 1 ppm acetone. To develop hierarchical designs for acetone sensing, Jing et al. [25] investigated the morphologies and microstructure of several ZnO nanomaterials, including ZnO flowers, disk pairs, and walnuts. As a result of having the highest degree of (001) facet exposure, largest BET-specific surface area, Schottky barrier height, and average pore size, ZnO flowers showed the best sensing performance in terms of responsiveness, selectivity, and stability for acetone, according to their analysis. Jia et al. [26] developed Fe_3_O_4_/rGO composites modified by Ag nanoparticles for acetone sensing, and their analysis found that the Ag0.1/Fe_3_O_4_/rGO sensor demonstrated an extremely high response of 35.81–50 ppm acetone at 220 °C. This performance was 2.5 times more effective than that of pure Fe_3_O_4_/rGO. The apparent improvement is attributable to the chemical and electrical sensitization effects of Ag nanoparticles, which generate additional oxygen species and active sites. Kim et al. [27] evaluated the acetone sensing performance of doped Co_3_O_4_ nanoparticles with SnO_2_ nanowires. Their results demonstrated a sevenfold increase in resistance to 50 ppm acetone gas compared to as-prepared SnO_2_. Increased variance in the p-n heterojunction’s surface depletion area was attributable to the observed enhancement.

The above studies suggest that numerous materials are being tested for their ability to detect acetone. This exemplifies the value of MGSs for acetone detection. The goal of this study is to investigate the feasibility of a vanadium oxide-based acetone sensor (V_2_O_5_). The sensitivity of vanadium oxide (V_2_O_5_) toward VOCs has been generating increasing attention due to its unique layered structure, low cost, and corrosion resistance, making it one of the most promising materials utilized in VOC detection [17]. Nonetheless, studies using V_2_O_5_-based sensors for acetone sensing are few in the literature. In this work, a V_2_O_5_ thin film was fabricated and its sensitivity toward acetone was evaluated. To increase the sensing capability of V_2_O_5_, its depletion layer was regulated by decoration with Au NPs and Ag NPs to form Ag NPs/V_2_O_5_ thin film/Au NPs heterostructures. The engineering of the V_2_O_5_ depletion layer with Au and Ag NPs leads to a considerable improvement in the sensing performance of V_2_O_5_ toward acetone [17].

## 2. Materials and Methods 

### 2.1. Synthesis of Ag NPs/V_2_O_5_ Thin Film/Au NPs

Figure 1 shows the steps used for fabricating Ag NPs/V_2_O_5_/Au NPs thin films. These steps can be divided into three main fabrication stages as follows.

#### 2.1.1. Fabrication of Ag NPs Thin Films

To fabricate Ag thin films, an Ag target was positioned in a DC magnetron sputtering system (Nano-Master, Austin, TX, USA). The substrate and the target were spaced 12 cm apart. The glass substrates and interdigitated electrodes (IDE) were cleaned for 20 min in an ultrasonic bath with ethyl alcohol, acetone, and DI water. The base pressure was held at around 2.8 × 10^−3^ torr, and the argon gas flow rate was maintained at 70 standard cubic centimeters (sccm) to maintain a working pressure of 3.1 × 10^−3^ torr. The sputtering period was 40 s, and the DC power was maintained at 50 W. To create well-dispersed Ag NPs, the Ag film was post-annealed in a tube furnace at 600 °C for 1 h in a nitrogen atmosphere.

#### 2.1.2. Fabrication of V_2_O_5_ Thin Film-Coated Ag NPs

A DC sputtering apparatus was used to sputter vanadium oxide on top of the Ag NPs layer. The only modifications to the deposition parameters in this stage were the addition of a reactive gas to the argon environment as well as the deposition power and time. A working pressure of 3.1 × 10^−3^ torr and a deposition power of 50 W were maintained for an hour by maintaining argon and oxygen gas flow rates at 70 sccm (30 O_2_, 40 Ar).

#### 2.1.3. Fabrication of Ag NPs/V_2_O_5_ Thin Film/Au NPs

The Ag NPs/V_2_O_5_ thin films were subsequently coated with an Au layer using a DC sputtering technique. Similar to the Ag target, the 99.99 percent pure Au target was pre-sputtered for 3 min at 40 W DC power. The working pressure was maintained at around 3.1 × 10^−3^ torr by maintaining the flow rate of argon gas at 70 sccm. The Au film was then transformed into Au NPs by heating the as-fabricated Ag NPs/V_2_O_5_/Au NPs films in a tube furnace at 600 °C.

### 2.2. Characterization Techniques

The fabricated films were characterized by using various techniques to study their morphology, composition, and microstructure. Powder X-ray diffraction (XRD, Rigaku MiniFlex, Austin, TX, USA) with Cu K radiation (=0.154178 nm) at 30 kV and 15 mA was used to examine the crystal structures of the produced films. The instrument that was used to investigate the vibration of the molecules was a Thermo Scientific Raman spectrometer (Thermo Scientific™ DXR3 Raman, Waltham, MA, USA) with an excitation wavelength of 455 nm, laser of power 6 MW, and resolution of 0.5 cm^−1^, and 20 scans were acquired between 100 and 2000 cm^−1^. A UV/Vis spectrophotometer (Jasco V-570, Tokyo, Japan) was used to measure optical transmittance in the wavelength range of 200–700 nm. The sample morphology characterization was conducted using a CSPM4000 Atomic Force Microscope (AFM, Nanosurf Easyscan, Liestal, Switzerland), in a contact mode and the frequency of the scanning was 2.0 Hz. 

### 2.3. Gas Sensing Measurements 

To increase the stability of the sensors, they were aged at 200 °C for 10 h. At various operating temperatures, the gas-detecting properties were evaluated for various acetone concentrations. A semiconductor device analyzer (SDA; Keysight B1500A) was used to determine how the electrical resistance changed in the presence of acetone (Rg) and air (Ro)

The sensitivity/response (S) of the sensors is given as follows:(1)$$S\left(\%\right)=\frac{\Delta R}{R_{0}}=\frac{R_{0}-R_{g}}{R_{0}}$$

## 3. Results and Discussion

### 3.1. Materials Characterization

Figure 2 shows the XRD patterns of the Ag thin film, Ag NPs, Ag NPs/V_2_O_5_, and Ag NPs/V_2_O_5_/Au NPs samples. The Ag thin film shows a diffraction peak at 38.19°, which corresponds to the (111) crystal plane with 0.52 full width at half maximum (FWHM), whereas Ag NPs display a diffraction peak at 38.4° and the FWMH increased to 6.69. In general, the width of a material phase is related to the average crystallite size of that material [18]. Decreasing particle size causes the widening of the peaks, which reflects the influence of the experimental circumstances on the nucleation and development of the crystal nuclei [19]. It is important to note that the (111) plane is the most stable in the fcc structure and has the highest surface energy. Furthermore, a weak XRD peak was also observed at 44.3° for both the Ag thin films and Ag NPs corresponding to the (100) plane [28]. This plane has the lowest surface energy and is the most inert. After depositing the V_2_O_5_ on the Ag NPs, only the above-mentioned diffraction peaks related to the Ag NPs were observed. The lack of diffraction peaks corresponding to V_2_O_5_ could be due to the amorphous nature of the materials or due to the limitation of the XRD system. When the Au was added to the Ag NPs and V_2_O_5_, there is no noticeable phase change, however, a significant improvement in the crystallinity of the material was observed.

To learn more about the vibrational modes of molecules and materials, Raman spectroscopy is frequently employed. The method relies on the frequency shift caused by the interaction of the scattered light with the vibrational modes of the sample during inelastic scattering The technique can be used for the identification and quantification of organic and inorganic chemicals, polymers, and minerals. It can also provide information about the structure, morphology, and orientation of materials [29,30,31,32,33,34]. Figure 3 shows the Raman spectra of the Ag NPs, V_2_O_5,_ Ag NPs/V_2_O_5,_ and Ag NPs/V_2_O_5_/Au samples.

A peak at around 400 cm^−1^ in the Raman spectrum of silver nanoparticles is often related to the transverse acoustic (TA) mode of the nanoparticles. The TA mode corresponds to the collective vibrations of silver atoms in the nanoparticle that are perpendicular to the direction of light propagation. The frequency of the TA mode is dependent on the size and structure of the nanoparticles, while the peak intensity offers information on the amount of silver atoms present in the nanoparticle [35].

In addition to the TA mode, there is another peak in the Raman spectrum close to 400 cm^−1^ that corresponds to other vibrational modes of the nanoparticles adsorbing on the nanoparticle surface. For instance, the presence of a peak at around 410 cm^−1^ has been linked to the stretching mode of sulfur atoms in thiol molecules adsorbed on silver nanoparticles’ surfaces. Another weak signal was observed at 600 cm^−1^ which has been associated with the breathing mode of the silver nanoparticle lattice. A peak found at 800 cm^−1^ could be associated with the breathing mode of silver nanoparticles This mode is connected to the nanoparticle’s overall contraction and expansion of the nanoparticles as a whole. Typically, a peak at around 1050 cm^−1^ is connected with the vibrational modes of the silver atoms in nanoparticles [36,37,38,39]. This peak is often assigned to the transverse optical (TO) mode, which corresponds to the collective vibrations of silver atoms in the nanoparticle parallel to the direction of light transmission.

For the V_2_O_5_ spectrum, weak signals were observed at around 390 cm^−1^, 610 cm^−1^, and 800 cm^−1^, which could be attributed to V-O, V-O-V, and O-V-O bending modes, respectively. In addition, another weak signal was observed at 1000 cm^−1^, which may correspond to the stretching vibrations of the V-O bonds in the VO_4_ tetrahedra [36,40,41,42]. Depositing a V_2_O_5_ thin layer over the Ag NPs does not result in any new peaks. There is an enhancement in the signal that may be attributed to a phenomenon referred to as surface-enhanced Raman spectroscopy. There is a broadening of the main peak that is also reflected in the Raman spectra of Ag NPs and V_2_O_5_. However, after adding the Au NPs on top of the V_2_O_5_ layer, it has observed that the Au has almost a similar peak position as in Ag NPs, so they overlap with the Ag NPs peaks and made the peaks look sharper.

Ultraviolet–visible (UV-Vis) spectroscopy was used to characterize the fabricated Ag thin film, Ag NPs, V_2_O_5_ thin film, Ag NP/V_2_O_5,_ and Ag NP/V_2_O_5_/Au samples (Figure 4). It is observed that there is a peak at 330 nm for Ag and after this peak, there is a sharp decrease in the transmittance, which reached up to 10% at 700 nm. The peak that was present in the Ag thin film disappeared after annealing, and the percentage of transmittance increased continuously from 40% at 200 nm to 65% at 700 nm, rather than dropping off sharply as before. This observation was due to the annealing process, which leaves some pores in the Ag thin films, which ensures better and improved transmittance. Conversely, the transmittance of V_2_O_5_ increases as the wavelength increases, and at 700 nm the transmittance reaches around 90%. After depositing the V_2_O_5_ thin film on Ag NPs, the transmittance continues to increase from 200 nm to 700 nm, with the percentage of transmitted light increasing from 13% to 63%, respectively. When the Au was deposited on the Ag NPs/V_2_O_5_ sample, it was observed that the transmittance was lower than that of V_2_O_5_, Ag NPs, and Ag NPs/V_2_O_5_. This is expected because the Au NPs act as an additional layer limiting the transmittance of light.

Atomic Force Microscopy (AFM) was used to reveal the surface structure modifications with composition change after depositing each layer. First, the surface appeared to be flat after the deposition of the Ag layer as shown in Figure 5a. Then after annealing the sample at 600 °C, Ag particles start grouping to form Ag NPs, which can be represented as peaks where the NPs are located and valleys where the atoms have moved from, as shown in Figure 5b. Then after the deposition of the V_2_O_5_ layer, V_2_O_5_ particles filled the holes and valleys caused by the annealing of the Ag layer and the surface appeared to be flat and smooth to some extent as shown in Figure 5c. Finally, after the deposition of the third Au layer and annealing the sample at 600 °C, we obtained Au NPs at the surface, which are shown in Figure 5d as sharp peaks and holes. More importantly, there is a significant increase in the surface area of the Ag NPs/V_2_O_5_ thin film/Au NPs after annealing at 600 °C.

### 3.2. Gas Sensing Properties

To activate gas detection, chemoresistive gas sensors normally require activation energy, which is frequently accomplished by an annealing process dictated by the working temperature. Because the sensitivity of gas sensors is significantly dependent on temperature, determining the temperature at which the sensor performs optimally is a critical task during the evaluation of chemoresistive gas sensors. Figure 6 shows the response curves of the V_2_O_5_, Ag NPs/V_2_O_5_ thin film, and Ag NPs/V_2_O_5_ thin film/Au NPs sensors exposed to 50 ppm acetone gas as a function of the working temperature. As can be observed, the response of the sensor is about 41% at room temperature. For breath analyzers lower concentrations of acetone are required, and the sensor’s response would be limited at room temperature for low concentrations. However, at high temperatures, the sensor can function as a breath analyzer even at very low concentrations of acetone. The best sensing response was obtained for Ag NPs/V_2_O_5_ thin film/Au NPs which were evaluated at different working temperatures ranging from room temperature 30 °C to 350 °C as shown in Figure 6. The optimal working temperature was 150 °C, as demonstrated by the fact that the gas response rises with operating temperature up to 150 °C and then begins to drop. Since Ag NPs/V_2_O_5_ thin film/Au NPs sensor has the best sensing performance among the different sensors, it was selected for studying acetone gas sensing. The dynamic response of the Ag NPs/V_2_O_5_ thin film/Au NPs sensor at the working temperature of 150 °C is shown in Figure 7. The response of the fabricated sensor is consistent with the amount of acetone passing in and out. The Ag NPs/V_2_O_5_ thin film/Au NPs resistance with respect to time was obtained for different concentrations of acetone (2, 5, 10, and 20 ppm) at 150 °C. As can be seen, the sensor resistance quickly decreases as the acetone concentration increases. The gas response of the Ag NPs/V_2_O_5_ thin film/Au NPs sensor toward 2 ppm is about 30%. With increasing acetone concentration to 20 ppm, the response of the Ag NPs/V_2_O_5_ thin film/Au NPs sensor increases to 84% at 150 °C. However, selectivity is an important factor for gas sensor applications, especially when examining complex gas mixtures, and the selectivity results in this work indicate that the sensor exhibited poor selectivity. Further work is needed to improve the selectivity of the sensor, either through doping V_2_O_5_ or by coupling V_2_O_5_ with other sensing materials.

Sensor reliability is measured by its repeatability. The repeatability of the Ag NPs/V_2_O_5_ thin film/Au NPs was evaluated by testing 20 ppm acetone at 150 °C for five exposure/recovery cycles, and the response-recovery curve is shown in Figure 8. It was noted that the fabricated sensor preserves its initial response amplitude with little or no decrease over five consecutive sensing tests toward 20 ppm acetone.

To establish the long-time stability of the sensor, the gas response was assessed between 2 to 20 ppm acetone at its optimal temperature over a period of 45 days and the results obtained are shown in Figure 9. The sensor demonstrates excellent stability.

### 3.3. Sensing Mechanism

The operation of the fabricated chemoresistive gas sensor depends on a change in electrical resistance and its sensing mechanism is shown in Figure 10. The V_2_O_5_ surface absorbed some of the oxygen contained in the atmosphere and as a result of the oxygen absorption, an ionization process is initiated which produces O^−^_2_, O^−^, and O^−2^ oxygen atoms. The working temperature determines the type of oxygen ions that will manifest [43].

The electrons move from the V_2_O_5_ layer creating what is referred to as an electron depletion region at the contact surface with O_2_. In the depletion region, the availability of the electrons is lower compared to the other regions within the same metal oxide [44,45]. The lack of electrons at the metal oxide surfaces results in upward surface band banding, which then causes the sensor resistance to increase [46,47]. When the depletion layer is exposed to acetone gas (a reducing gas), the acetone reacts with the surface oxygen ions. This process returns electrons to the conduction band, reducing the electron depletion zone and, as a consequence, lowering resistance, as illustrated in Figure 10b [45,46,47,48]. The presence of Au NPs on top of the V_2_O_5_ helps to increase the number of electrons on the surface, due to the difference in work function between the Au and V_2_O_5_, which makes the depletion region thicker, thereby resulting in a significant improvement in the sensor resistivity (Figure 10c,d) [49]. The addition of the Ag NPs under the V_2_O_5_ layer also produces a depletion region beneath the V_2_O_5_ due to the difference in work function between the Ag and V_2_O_5_. The work function of Ag is less than that of V_2_O_5_, which in turn leads to the inducement of electron drift to V_2_O_5_ from Ag NPs. In addition, the work function of Au is greater than that of V_2_O_5_, therefore there will be an additional flow from V_2_O_5_ to Au until the Fermi level of the three elements achieves equilibrium. When equilibrium is attained, the creation of a thick depletion zone in the Ag and V_2_O_5_ with increased electron density at the surface is probable, and this kind of surface rearrangement promotes the adsorption of oxygen molecules.

When the acetone gas is introduced to the sensor surface (Figure 10e), which contains O_2_ gas, the acetone will react with the O_2_ causing the depletion layer to become thinner (Figure 10f). The following chemical reaction equations explain the sensor response to the acetone gas:CH_3_COCH_3_ (gas) + O^−^_2_ → H_2_O (gas) + CO_2_ (gas) + 2e^−^(2)

The thickness of the depletion layer (W) is given by the equation below:W = L_d_(qV_s_/kT)^1/2^(3) where L_d_, q, K, Vs, and T are the Debye length, elementary charge, Boltzmann constant, band bending, and operating temperature, respectively. The significant improvement observed in the fabricated sensor (Ag NPs/V_2_O_5_ thin film/Au NPs) can be attributed to the modulation of the depletion layer, which is affected by several factors, such as the shell layer, doping, polymorphic forms, decoration (layers other than the V_2_O_5_), and the preparation methods [49,50,51]. Through the spillover effect, Au NPs are a well-known chemical sensitizer capable of providing more adsorbed oxygen molecules onto the Ag NPs/V_2_O_5_ thin film/Au NPs thin film. The greater the amount of adsorbed oxygen species, the more reaction sites for acetone available on the Ag NPs/V_2_O_5_ thin film/Au NPs surface, and thus the higher the response. In addition, the double activation of the V_2_O_5_ layer resulted in a thicker depletion region, resulting in a significant enhancement in acetone sensing. Furthermore, increasing the surface area where the oxygen gas can react and form a depletion region will lead to improving the sensitivity. In the Ag NPs/V_2_O_5_ thin film/Au NPs, the fabricated sensor has a better surface area compared with other thin films and hence it exhibited increased performance. Figure 10 shows the mechanism for the three types of sensors (V_2_O_5_, Ag/V_2_O_5_ thin film, and Ag NPs/V_2_O_5_ thin film/Au NPs).

## 4. Conclusions

Using a combination of DC/RF sputtering and post-annealing as synthesis techniques, we successfully fabricated the following nanomaterial: V_2_O_5_, Ag/V_2_O_5_, and Ag NPs/V_2_O_5_/Au NPs, and subsequently examined their responses toward acetone detection as a potential volatile biomarker. The structural, optical, morphological, and vibration modes of these nanomaterials were investigated using XRD, UV-vis spectroscopy, AFM, and Raman spectroscopy, respectively. The findings from these characterizations showed consistency with the existing literature.

The fabricated Ag NPs/V_2_O_5_/Au NPs sensor demonstrates the highest response (about 96%) among the samples that were fabricated. This is in contrast to the Ag/V_2_O_5_ and V_2_O_5_ samples, which exhibited responses of 62% and 23%, respectively, for 50 ppm acetone. It was determined that 150 °C is the optimal temperature for the Ag NPs/V_2_O_5_ thin film/Au NPs sensor, and the sensor demonstrated a long-term stability of 45 days for 20 ppm acetone. The engineering of the layer depletion that was carried out by dopant decoration (layers other than V_2_O_5_) as well as the preparation processes are responsible for the improved sensor performance observed in the Ag NPs/V_2_O_5_ thin film/Au NPs material.

## Figures and Tables

**Figure 1 nanomaterials-13-01372-f001:**
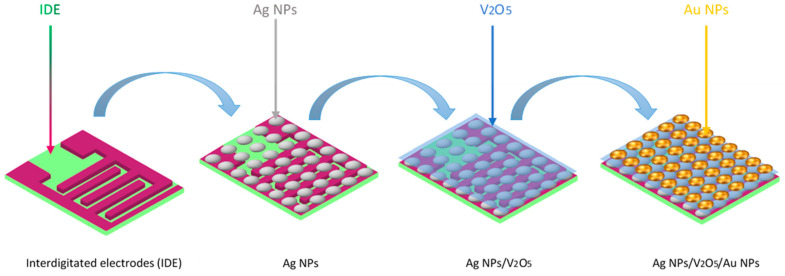
Schematic diagram of the preparation process of Ag NPs/V_2_O_5_ thin film/Au NPs.

**Figure 2 nanomaterials-13-01372-f002:**
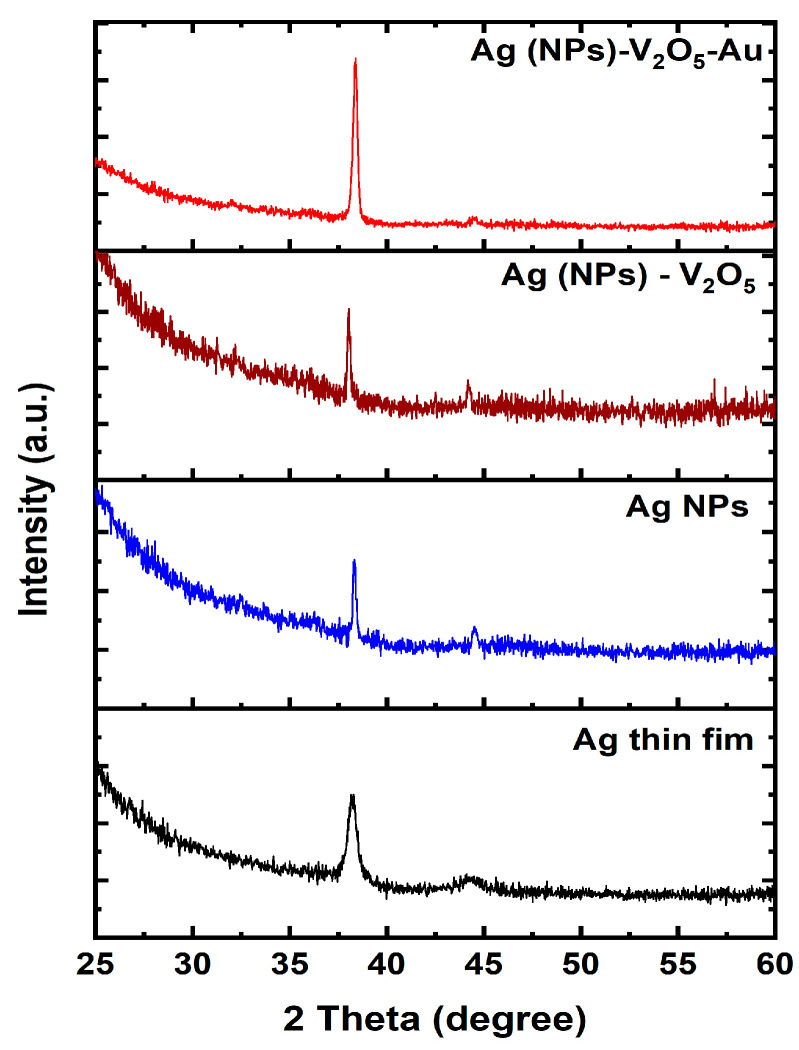
XRD patterns of Ag NPs/V_2_O_5_/Au NPs, Ag NPs/V_2_O_5_, Ag NPs, and Ag NPs thin film.

**Figure 3 nanomaterials-13-01372-f003:**
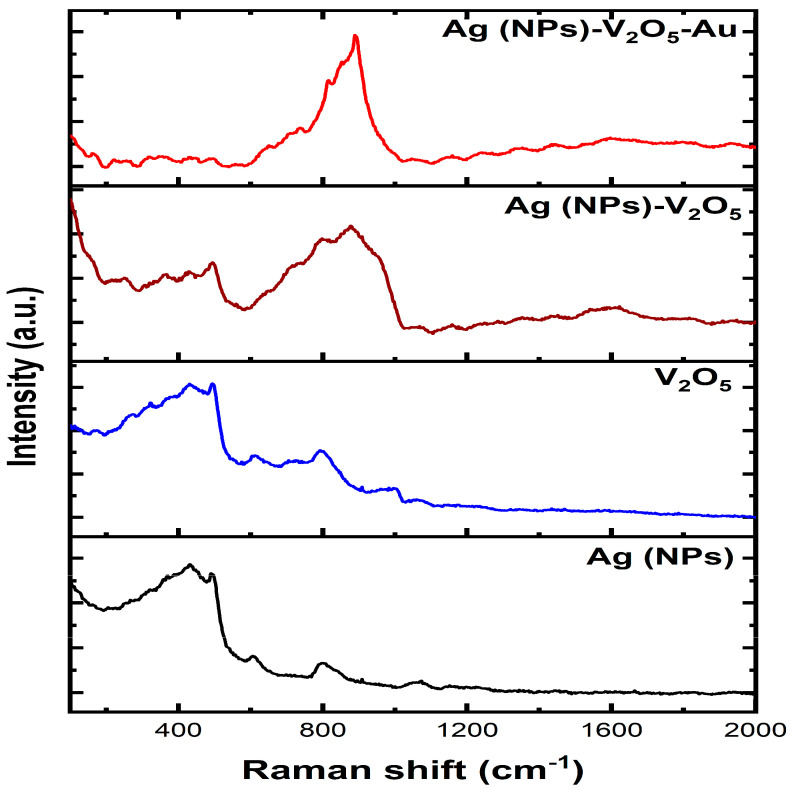
Raman of as-deposited Ag thin film, Ag NPs, Ag NPs/V_2_O_5_ thin film, and Ag NPs/V_2_O_5_ thin film/Au NPs.

**Figure 4 nanomaterials-13-01372-f004:**
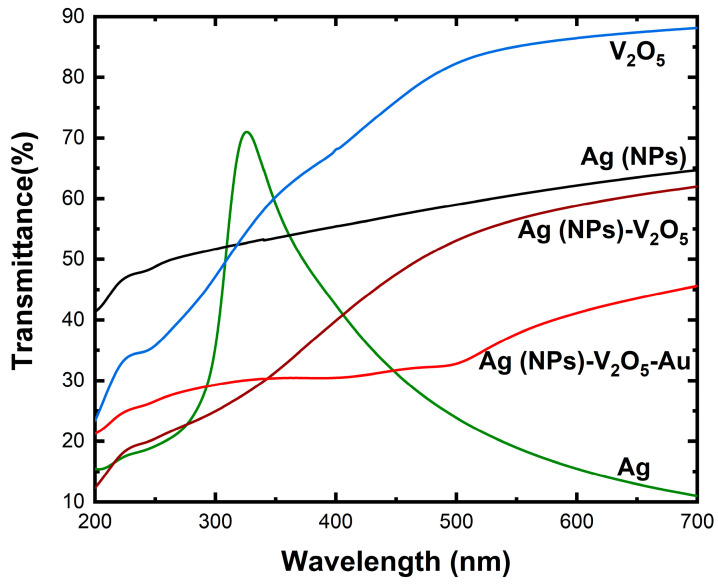
UV-Vis spectra of Ag thin film, Ag (NPs), V_2_O_5_ thin film, Ag NPs/V_2_O_5_, and Ag NPs/V_2_O_5_ thin film/Au NPs.

**Figure 5 nanomaterials-13-01372-f005:**
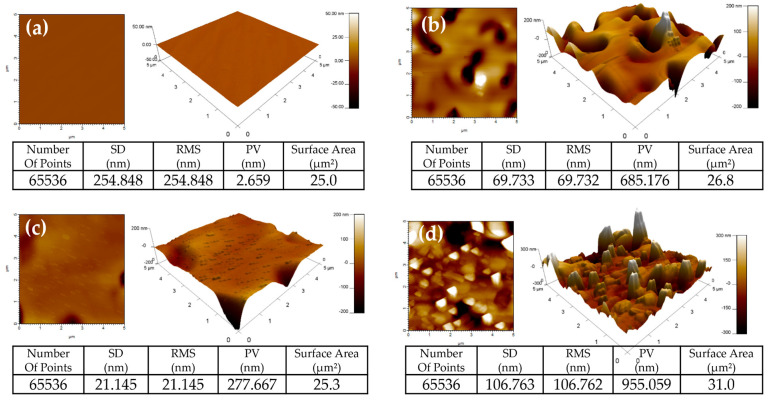
2D and 3D AFM morphology images of (**a**) Ag layer, (**b**) Ag layer after annealing it at 600 °C (Ag NPs), (**c**) Ag NPs/V_2_O_5_ thin film, (**d**) Ag NPs/V_2_O_5_ thin film/Au NPs after annealing at 600 °C.

**Figure 6 nanomaterials-13-01372-f006:**
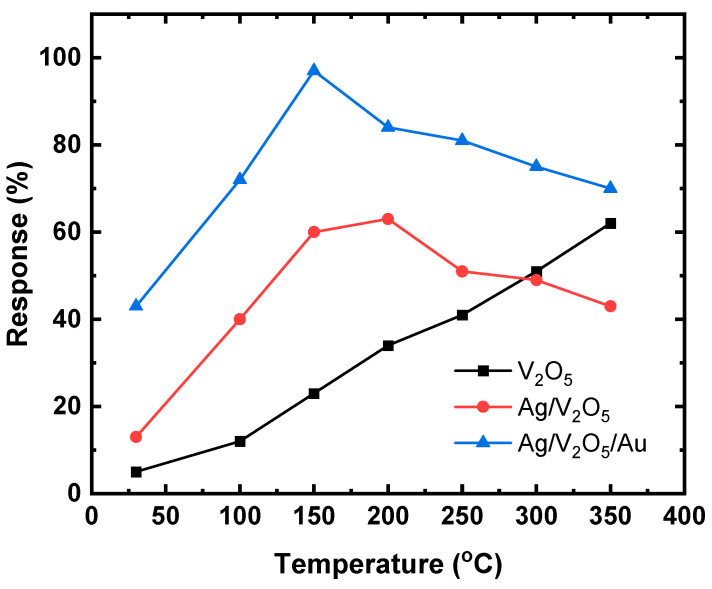
Response curves of the V_2_O_5_, Ag NPs/V_2_O_5_ thin film, Ag NPs/V_2_O_5_ thin film/Au NPs sensors exposure to 50 ppm acetone gas as a function of working temperature.

**Figure 7 nanomaterials-13-01372-f007:**
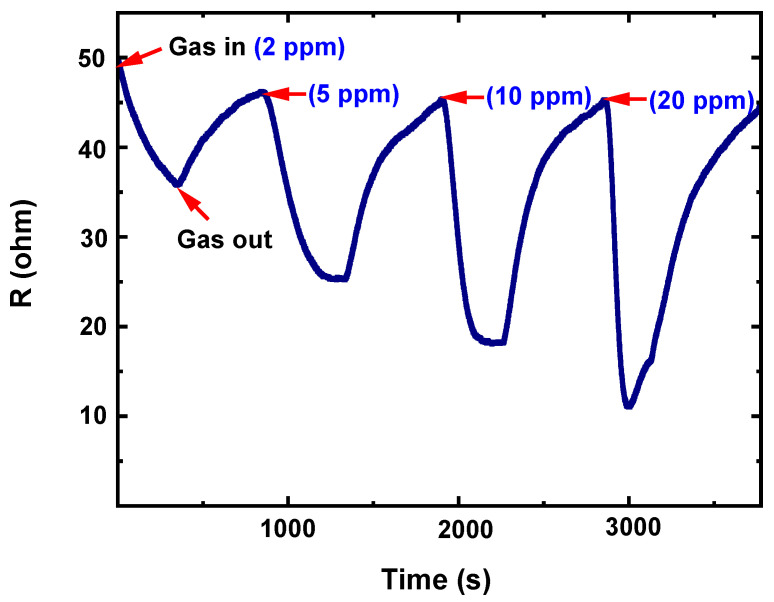
Dynamic resistance of the Ag NPs/V_2_O_5_ thin film/Au NPs sensor exposure to acetone gas with different concentrations ranging from 2 ppm to 20 ppm at 150 °C.

**Figure 8 nanomaterials-13-01372-f008:**
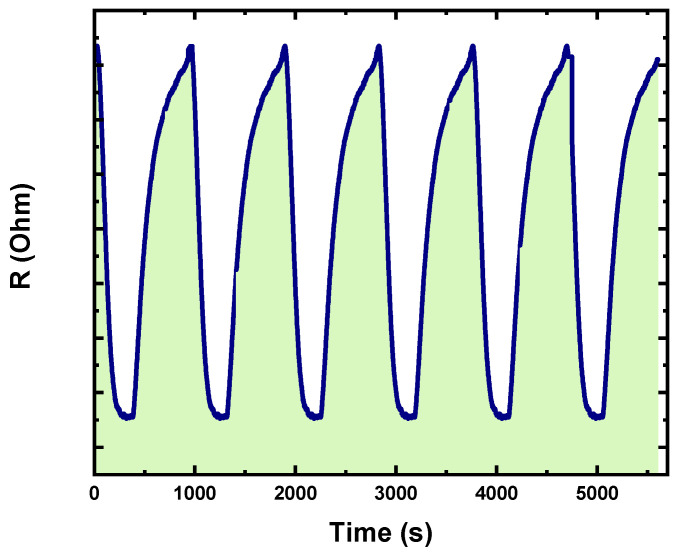
Repeatability test of the Ag NPs/V_2_O_5_ thin film/Au NPs across eight repeats.

**Figure 9 nanomaterials-13-01372-f009:**
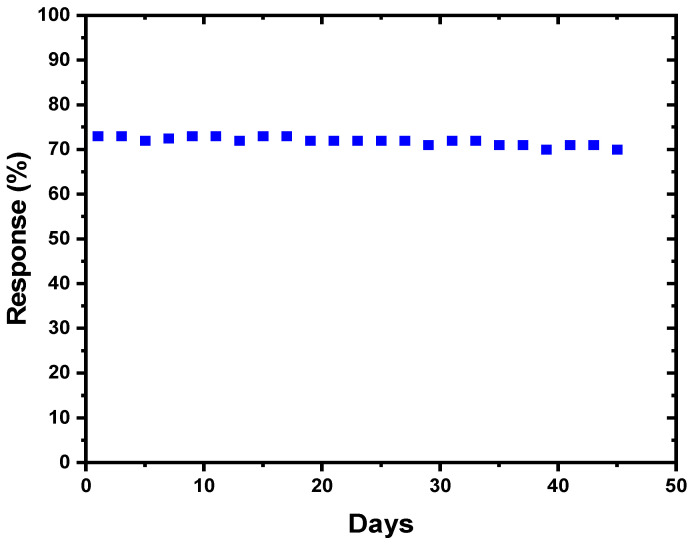
Long-term stability of the Ag NPs/V_2_O_5_ thin film/Au NPs exposed to 20 ppm acetone at 150 °C.

**Figure 10 nanomaterials-13-01372-f010:**
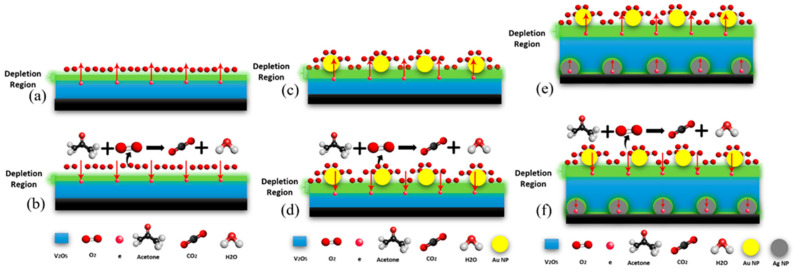
Mechanism illustration for acetone gas sensing in the presence of V_2_O_5_ (**a**,**b**), Ag/V_2_O_5_ thin film (**c**,**d**), and Ag NPs/V_2_O_5_ thin film/Au NPs (**e**,**f**).

## Data Availability

The data presented in this study are available on request from the corresponding author.

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
