# Peer review of "Regulating the Electron Depletion Layer of Au/V2O5/Ag Thin Film Sensor for Breath Acetone as Potential Volatile Biomarker"

_nanomaterials, 2023, doi:10.3390/nano13081372_

Round 1

Reviewer 1 Report

In manuscript, the authors developed a breath acetone sensor based on Ag NPs/ V2O5 thin film/Au NPs by combining DC/RF sputtering and post-annealing as synthesis methods. The as-produced sensor has high sensitivity, which is nearly twice and four times greater than the sensitivity of Ag NPs/ V2O5 and pristine V2O5, respectively. The mechanism of the increase in sensitivity has been discussed. Specific comments,

(1)   The wavenumber of Figure 3 should be shown in from high to low.

(2)   The Dynamic resistance of the sensor exposure to 50 ppm acetone was not found in the manuscript.

(3)   The inter batch differences of sensors should be evaluated.

(4)   The term of ‘UV-Vis spectroscopy’ in the Figure 4 should be corrected as ‘UV-Vis spectra’.

(5)   Is the sensitivity of as-developed sensor dependent on the surface intensities of Ag NPs and/or Au NPs?

Reviewer 2 Report

The manuscript, “Regulating the electron depletion layer of Au/V2O5/Ag thin film sensor for breath acetone as potential volatile biomarker” by Alghamdi et al are designed and fabricated to a novel breath acetone sensor. A variety of techniques were used to characterize the obtained material, which demonstrated improved sensitivity and selectivity. Also, I believe that this work is acceptable for publication in nanomaterials with the following minor improvements:

1.      In Figure 2 of the revised manuscript, the authors should properly discuss and describe the crystal planes in the XRD patterns.

2.      The Raman analysis should be discussed in detail by authors with the appropriate citations, as discussed in previous published papers.

3.      The authors must reevaluate the UV-Vis spectrum and clearly explain their findings in the revised manuscript.

4.      Some of the important references need to cite in the revised manuscript.  10.1016/j.apsusc.2021.149765; 10.1039/C6NJ04030F

5.      The gas sensing mechanism and conclusion section should be discussed in detail by the authors.

Reviewer 3 Report

In this manuscript, the authors prepare a sensor made of AgNPs/ V2O5 thin film/AuNPs by combining DC/RF sputtering and post-annealing. The sensor is targeted for breath acetone sensing. The authors characterize the material using X-ray diffraction, Raman and UV-Vis Spectroscopy, and AFM. The authors showed a sensitivity to 50 ppm acetone of the AgNPs/ V2O5 thin film/AuNPs is 96%. Below are some questions and comments.

1. How uniformly dispersed are the AuNPs on the surface when laminating V2Oin between?

2. From the AFM morphology in Figure 5, it appears AgNPs/ V2O5 thin film/AuNPs has large surface area of AgNPs. Is surface area another reason of enhanced sensitivity? More comments are needed here.

3. As shown in Figure 6, the response of AgNPs/ V2O5 thin film/AuNPs increase more than two fold from room temperature to 150 degC. However, room temperature would be a more practical application condition. How sensitive is the sensor under room temperature? More comments are needed here.

4. Does the sensor response linearly to different concentration of acetone gas? A characterization experiment is needed.

5. The exhaled gas from patient also contains humidity. How does the sensor respond under humid conditions? More experiments are needed.

6. How fast is the response time? A "rise time" characterization of the sensor is needed.

7. The authors only used acetone gas as a test. How selective is the sensor? What if another type of organic solves is supplied to the sensor. What's the response compare to acetone gas? Control experiments are needed.

Round 2

Reviewer 1 Report

The revised manuscript could be published as it.

Author Response

Comment from the Reviewer: 

The revised manuscript could be published as it.

Response: 

We are very grateful to the reviewer for his/her constructive suggestions and for his/her proposed corrections to improve our manuscript. 

Reviewer 3 Report

The authors need to mention in the manuscript about the low sensitivity at room temperature, and the poor selectivity over different molecules, and how they plan to improve these in the manuscript before publication can be recommended.

Author Response

Reviewer's comment

The authors need to mention in the manuscript about the low sensitivity at room temperature, and the poor selectivity over different molecules, and how they plan to improve these in the manuscript before publication can be recommended.

Response:

We thank the reviewer for his/her recommendation. we have revised the manuscript based on his/her recommendation.